



# Explainable AI for Knowledge Acquisition in Hydrochemical Time Series V1.0.0

Michael C. Thrun[1], Alfred Ultsch[1], Lutz Breuer[2]

[1]Databionics Research Group, University of Marburg, Germany
[2]Institute for Landscape Ecology and Resources Management (ILR), Justus Liebig University Giessen

*Correspondence to*: Michael C. Thrun: (mthrun@mathematik.uni-marburg.de)

**Abstract.** The understanding of water quality and its underlying processes is important for the protection of aquatic environments. Here an explainable AI (XAI) based multivariate time series analytical framework is applied on high-frequency water quality measurements including nitrate and electrical conductivity and twelve other environmental

parameters. The relationships between water quality and the environmental parameters are investigated by a cluster analysis which does not depend on prior knowledge about data structure. The cluster analysis is designed to find similar days within a cluster and dissimilar days between clusters. This allows for the data-driven choice of a distance measure. Using a swarm based AI system, the resulting cluster define three states of water bodies, which can be visualized by a topographic map of high-dimensional structures. These structures are explained by rules extracted from decision trees.

The rules generated by the XAI system improve the understanding of aquatic environments. The model description presented here allows to extract meaningful, useful, and new knowledge from multivariate time series.

**Keywords**: Data-driven Knowledge Acquisition; Cluster Analysis; Swarm Intelligence; Machine Learning System; High-Dimensional Data Visualization

**1 Introduction**

Human activities modify the global nitrogen cycle, particularly through agriculture. These practices have unintended consequences; for example, terrestrial nitrate losses to streams and estuaries can impact aquatic life (Durand et al., 2011). A greater understanding of the variability in water quality and its underlying processes can improve the evaluation of the state of water bodies and lead to better recommendations for appropriate and efficient management

practices (Cirmo and McDonnell, 1997).

Accordingly, the objective here is to describe the water quality in terms of nitrate ($NO_3$) and electrical conductivity (EC) in the Schwingbach catchment (Germany) using environmental variables typically related to chemical water quality. Electrical conductivity is a measure that reflects water quality as a whole because it indicates the number of ions dissolved in the water. $NO_3$ in water bodies is partially responsible for the phenomenon of eutrophication (Diaz,

2001). Eutrophication occurs when an excess of nutrients (including $NO_3$) leads to the uncontrollable growth of aquatic plant life, followed by the depletion of the dissolved oxygen (Diaz, 2001; Howarth et al., 1996). For decades, water quality has mainly been measured through manual grab sampling of water samples and subsequent chemical analysis in the laboratory. Due to limited resources, high-resolution measurements on the order of days, hours or even minutes were not available for a long time. With the advancement of deployable, *in situ* measuring techniques, such as UV



spectrometry, a new era of field monitoring has been established (Rode et al., 2016). However, we are still lacking methodological approaches with which to analyze the resulting large datasets (Aubert and Breuer, 2016; Aubert et al., 2016).

Typically, time series clustering is raw-data based, feature based or model based (Liao, 2005). Often, either the Euclidean metric or Kullback-Leibler dissimilarity are selected, and cluster analysis is performed via medoid or

agglomerative methods of conventional clustering algorithms (Liao, 2005). Adapting conventional clustering algorithms results in a sensitivity to outliers and noise (Ferreira and Zhao, 2016; Ma et al., 2019). Such approaches also imply that the relevant cluster structures are spherical or of other specific structures because a global optimization function has to be used (Thrun, 2018). Then, evaluation is commonly performed on the basis of within-cluster variance (Liao, 2005). Other common approaches to the evaluation of time series are shape based, meaning that the shapes of

two time series are matched according to specific dissimilarity measures (Aghabozorgi et al., 2015). The approaches have in common that they optimize a global objective function which defines the cluster structures, and do not investigate if cluster structures exist. Contrary to these approaches this work applies an alternative approach to the optimization of a global objective function in the task of clustering called Databionic swarm (DBS) (Thrun and Ultsch, 2020a). In contrast to most other clustering algorithms, the topographic map visualization of DBS identifies if the

clustering of the data is meaningless in the sense that the data contains no clusters (Thrun and Ultsch, 2020a). Outliers can be interactively marked in the visualization after the automated clustering process in the case that they are not recognized sufficiently in the automatic clustering process (Thrun and Ultsch, 2020a).

The main contribution of this paper is the proposition of a machine learning system that reveals cluster structures in time series with a solely data-driven approach. In this context, data-driven means that the authors aim to refrain from

making explicit or implicit assumptions about the data, existence and type of cluster structures. The found cluster structures are verified with independent approaches. Furthermore, knowledge acquisition is applied to describe the clusters, leading to the discovery of new knowledge. The procedure is introduced using a dataset from the Schwingbach catchment published earlier by Aubert et al. (Aubert et al., 2016). However, Aubert et al. used a temporal high-frequency analysis. In comparison, this work focuses on the average daily measures for each variable, resulting in a

low-frequency analysis.

Overall, this work shows how to search for days with similar behavior by using a swarm-based clustering approach. The goal is to explain similar environmental, and in particular water quality, situations by human understandable rules because $NO_3$ stream concentrations "integrate" many processes varying in space and time (Pellerin et al., 2009). Finally, we use these rules to describe the clusters to predict future $NO_3$ and EC values.

## 2 Material and Methods

The analytical procedure for the explainable AI for knowledge acquisition is presented in Fig. 1. The methods sections are organized in six steps as illustrated in the titles of Fig. 1.





### 2.1 Step I: Data Preprocessing

The dataset contains 32,196 data points for 14 different variables. In table 1, a detailed description of the variables is

provided. Technical information on the analytical procedures and experimental design of the field work are outlined by
(Aubert et al., 2016).

All data were recorded at high frequency (15-min intervals) and span a total of 2 years of measurements, focusing on
the summer period between April and October. Four percent of the data are missing. For each day, the measurements
were aggregated by the mean of all available measurements for that day. Then, missing values (i.e., days) were

interpolated using the seven-nearest-neighbors approach. The variables q13 and q18 were log transformed. All
variables, with the exception of rainfall, were normalized to values between zero and one through a robust normalization
procedure (Milligan and Cooper, 1988) improved by (Thrun, 2018). The discharges correlated linearly with each other
(r=0.95, p(S=347,270, N=351) <0.001), and q13 was therefore excluded from the analysis. The air temperatures Wt13
and Wt18 also correlated linearly (r=0.99, p(S=18,386, N=351) <0.001); hence, Wt13 was removed as well.

The outliers in the rainfall variable were detected via ABC analysis (Ultsch and Lötsch, 2015). ABC analysis is a
method used to compute precise limits to acquire subsets in skewed distributions by exploiting the mathematical
properties pertaining to the distribution. The data containing positive values are divided into three disjoint subsets, A,
B and C, with subset A comprising very profitable values, i.e., largest data values ("the important few"), subset B
comprising values where the yield equals the effort required to obtain it, and subset C comprising non profitable values,

i.e., the smallest datasets ("the trivial many"). The R package is available on CRAN (https://CRAN.R-
project.org/package=ABCanalysis). Then, rain was normalized with respect to the minimum value in group A. All
other points in group A were capped by defining the upper bound for rainfall. In supplementary Information A (SI A),
the probability density distributions of the 12 finally selected and transformed variables are visualized with mirrored-
density plots (Thrun et al., 2020). The mirrored-density plots (MD-plots) show that the range of a variable is

approximately between zero and one (SI. A, Fig. 10) because the normalization approach uses 1 and 99% quantiles
instead of maxima and minima and thus allows outliers to lie below zero or above one.

### 2.2 Step II: Distance Selection

The Hellinger distance measure is selected and the DBS method (Thrun, 2018; Thrun and Ultsch, 2020a) is applied in
three modules for cluster analysis. DBS is available in the R package 'DatabionicSwarm' on CRAN (https://CRAN.R-

project.org/package=DatabionicSwarm). This specific metric is chosen because clear multimodality is visible in the
probability density distribution. Several metrics were investigated using the R package 'parallelDist' (Eckert, 2018) on
CRAN (https://CRAN.R-project.org/package=parallelDist) and the MD-plot function (Thrun et al., 2020) in the R
package 'DataVisualizations' on CRAN (https://CRAN.R-project.org/package=DataVisualizations). The probability
density distribution is modeled with a Gaussian mixture model and verified visually and statistically as described in

(Ultsch et al., 2015) with the R package 'AdaptGauss' on CRAN (https://CRAN.R-project.org/package=AdaptGauss).


### 2.3 Step III: Clsuter Analysis

In the first step of the DBS method, the high-dimensional dataset is projected on the two-dimensional plane with a swarm-based projection method called Pswarm (Thrun, 2018; Thrun and Ultsch, 2020a). By exploiting concepts of

self-organization and emergence, swarm intelligence and game theory, this projection method is parameter free and nonlinear (Thrun, 2018; Thrun and Ultsch, 2020a). The swarm first adapts to global structures, and as time progresses, structure preservation shifts from global optimization to the preservation of local neighborhoods. Projections of this type are called focusing and usually require parameters to be set (c.f. (Ultsch, 1999; Van der Maaten and Hinton, 2008)) because this phase, which is also called the learning phase, requires an annealing scheme. The intelligent agents of

Pswarm, called DataBots (Ultsch, 2000), operate on a toroid grid, where positions are coded into polar coordinates to allow for the precise definition of their movement, neighborhood function and annealing scheme. The size of the grid and, in contrast to other focusing projection methods (e.g., (Demartines and Hérault, 1995; Ultsch et al., 2016; Van der Maaten and Hinton, 2008)), the annealing scheme are data driven. Therefore, this method does not require any parameters. During learning, each agent moves across the grid or stays in its current position in the search for the most

potent scent. Hence, agents search for other agents carrying data with the most similar features to themselves with a data-driven decreasing search radius (Thrun and Ultsch, 2020a). The movement of every agent is modeled using a game-theory approach, and the radius decreases only if a Nash equilibrium is found (Nash, 1950, 1951).

In the second step, the generalized U-matrix (Thrun, 2018; Ultsch and Thrun, 2017) is calculated on this projection using emergence through an unsupervised artificial neural network called a simplified (because parameter free)

emergent self-organizing map. The generalized U-matrix generates the visualization of a topographic map with hypsometric tints, which can be vividly described as a virtual 3D landscape with a specific color scale chosen with an algorithm defining the contour lines (Thrun et al., 2016). The topographic map addresses the central problem in clustering, i.e., the correct estimation of the number of clusters. It allows the assessment of the number of clusters by inspecting the 3D landscape.

The topographic maps correspond to high-dimensional distance and density structures. Hypsometric tints are surface colors that represent ranges of elevation. The contour lines are combined with a specific color scale. The specific colour scale is chosen to display various valleys, ridges, and basins: blue colours indicate small distances (sea level), green and brown colours indicate middle distances (low hills), and shades of white colours indicate vast distances (high mountains covered with snow and ice). Valleys and basins represent clusters, and the watersheds of hills and mountains

represent the borders between clusters. In this 3D landscape, the borders of the visualisation are cyclically connected with a periodicity (L,C).

Finally, in step 3, semi automated clustering of the visualization is applied by calculating the shortest paths (Dijkstra, 1959)) of the Delaunay graph between all projected points weighted with high-dimensional Hellinger distances. This is possible because it was shown that the U-matrix is an approximation of the abstract U-matrix(Lötsch and Ultsch,

2014), which is based on Voronoi cells. Voronoi cells define a Delaunay graph where the edges between every projected point are weighted by the high-dimensional distances of the corresponding data points.

The clustering approach itself involves two choices. For this dataset, the compact approach is used, where the two clusters with the minimal variance ($S$) are merged together until the number of clusters defined by the topographic map



is reached. The other approach for connected structures and a general discussed of cluster structures can be found in
(Thrun and Ultsch, 2020b). In praxis, the choice of this parameter can be evaluated in step IV (Fig. 1).

Let $c_r \subset I$ and $c_q \subset I$ be two clusters such that $r, q \in \{1, \ldots, k\}$ and $c_r \cap c_q = \{\}$ for $r \neq q$ and

$$\Delta Q(j, l) = \frac{k * p}{k + p} * D^*(j, l) \qquad (1)$$

where

| | |
|---|---|
| $(l, j)$ | the data points in the clusters be denoted by $j_i \in c_q$ and $l_i \in c_r$; |
| $k$ | the cardinality $\lvert c_q \rvert$ of the first set; |
| $p$ | the cardinality $\lvert c_r \rvert$ of the second set; |
| $D^*$ | the high-dimensional distance based on weighted shortest paths in the Delaunay graph; |
| $\Delta Q$ | the merging cost between two the clusters $c_r, c_q \subset I$ |

Then, the variance ($S$) between two clusters ($c_r$ and $c_k$) is defined as

$$S(c_r, c_k) = \sum_{i=1, j=!, j \neq i}^{k, p} \Delta Q(j, l) \qquad (2)$$

The ultrametric portion of the distance ((Murtagh, 2004)) can be visualized by a dendrogram allowing the alternative
selection of the number of clusters: Large changes in the fusion levels of the ultrametric portion of the distance indicate
the best cut.

The three modules, of the algorithm (projection, topographic map of structures, clustering) are called the Databionic
swarm (Thrun, 2018) and are available in R language on CRAN in the package 'DatabionicSwarm'

**2.4 Step IV Validation of Clustering**

The clustering is valid if mountains do not partition clusters indicated by colored points of the same color and colored
regions of points. Further, the clustering is verified by the heatmap of distances which are ordered by the clustering
(Wilkinson and Friendly, 2012) because this work searches for days with similar behavior.. A heatmap visualizes the
homogeneity of clusters and the heterogeneity of intercluster distances if the clustering is appropriate. The R package
'DataVisualizations' on CRAN is used (Thrun and Ultsch, 2018).

Ockham's razor states that if two models are applicable, the less complex one should be used (Sober, 1991). Therefore,
the authors try simpler models in order to evaluate if the complexity of DBS is necessary to derive the true cluster
structures of the data. A simpler projection approach assuming linear cluster structures and a simpler clustering
approach assuming ellipsoidal cluster structures is applied to the data. Moreover, spherical cluster structures are tested
with the Silhouette plot using the R package "DataVisualizations" on CRAN.

**2.5 Step V: Knowledge Acquisition**

Decision trees are supervised methods like the classification and regression tree (CART) (Breiman et al., 1984) or
globally optimal classification and regression trees (Grubinger et al., 2014). For that purpose, R package "rpart" on
CRAN (Therneau et al., 2018) and the package "evtree" (Grubinger et al., 2014) are applied. In this work, decision



trees are computed using DBS clustering and back-transformed data. The standardization had to be back transformed to provide SI units of measurement. Decision tree algorithms do not aim at understandable and meaningful explanations (Mörchen and Ultsch, 2007; Mörchen et al., 2005). Therefore a transformation into human-understandable rules is necessary (Mörchen and Ultsch, 2007; Mörchen et al., 2005). Here, rules are extracted from the decision tree by following each path from root to leaf and, if necessary, simplifying the rules manually.

### 2.6 Step VI: Knowledge Discovery (KD)


In KD context here, the question is whether the found cluster structures describe different states of $NO_3$ and EC. The hypothesis is that if the topographic map of the hydrochemical data reveals meaningful high-dimensional structures for similar days, then the classes should contain samples of different environmental states and be based on different processes. To verify this hypothesis, the mirrored-density plots (Thrun et al., 2020) of each class are used to show the dependence of the clusters on $NO_3$ and EC concentrations.


The Mirrored-Density plot (MD-plot) introduced in (Thrun et al., 2020) visualizes a density estimation in a similar way to the violin plot (Hintze and Nelson). The MD-plot uses for density estimation the Pareto density estimation (PDE) approach (Ultsch, 2005). It can be shown that comparable methods have difficulties in visualizing the probability density function in case of uniform, multimodal, skewed, and clipped data if density estimation parameters remain in a default setting (Thrun et al., 2020). In contrast, the MD plot is particularly designed to discover interesting structures in continuous features and can outperform conventional methods (Thrun et al., 2020). The MD plot does not require any adjustments of parameters of density estimation, which makes the usage compelling for non-experts. The MD-plot is available in the R package 'DataVisualizations' on CRAN (Thrun and Ultsch, 2018).


### 3 Results

An overview of the analysis is provided in Fig. 1. For clarity the rest of this chapter is subdivided into five sections: the first section consists of the selection of an appropriate distance metric and extracting the first hypothesis from the distribution of distances (3.1). The second section presents the Databionic swarm cluster analysis method (3.2). The next section validates the clustering (3.3). The fourth section is knowledge acquisition (3.4), and the last section is knowledge discovery (3.5).

### 3.1 Step II: Distance Selection


The Hellinger distance (Rao, 1995) in the R package 'parallelDist' on CRAN was chosen for cluster analysis because the distribution of distances is statistically not unimodal according to Hartigan's dip test (Hartigan and Hartigan, 1985) (with $p(D = 0.006385, N= 61425)<0.001$). The distance distribution can be modeled through Gaussian mixture model (GMM) using the expectation maximization (EM) algorithm (Dempster et al., 1977). The distance distribution and model isvisualized in Fig. 2. The QQ-plot verifies the GMM in Fig. 3. This serves as an indication of the existence of high intercluster distances (distances between different clusters) and outlier distances as well as small intracluster distances (distances within each cluster), meaning that a distance-based cluster structure can be found.




### 3.2 Step III: Cluster Analysis

Next, the topographic map with hypsometric tints generated with the R package 'DatabionicSwarm' from CRAN is
toroidal, meaning that the borders of the grid are cyclically connected with a periodicity defined by the size of the grid
of the projection of Pswarm. In Fig. 4, a cutout island of the topographic map is shown. Every point symbolizes a day.
The high-dimensional distances of the low-dimensional projected points are visualized. The topographic map shows
three valleys and basins indicating clusters and watersheds of hills and mountains shown by borderlines between
clusters. Thus, the number of clusters is equal to the number of valleys. The labels of the clusters are hereafter visualized
as the colors of the projected points in Fig. 4. In addition to the two main clusters (magenta and yellow points) and one
outlier cluster (black points), seven outliers can be identified as volcanoes or within the valleys which is indicated by
red arrows in Fig. 4.

### 3.3 Step IV: Validation of Cluster Analysis

, First, the authors created a heatmap in order to verify the DBS visualization and clustering. The heatmap shows intra-
versus intercluster distances ordered by each cluster (Fig. 5). Blue colors symbolize small distances, and yellow and
red colors represent large distances. The median intra-cluster distances of clusters 1, 2 and 3 are 0.24, 0.36 and 0.31,
respectively and are below the Bayes boundary of the GMM in Fig. 1 of 0.39. The average intercluster distance is 0.48
and above the Bayes boundary of 0.39. These results indicate that the intracluster distances are smaller than the
intercluster distances. This means that days within each cluster are evidently more similar to one another than days
between clusters.

To check for a possibility of a simpler model, a linear projection by the method projection pursuit (Hofmeyr and
Pavlidis, 2015) using a clusterability index of variance and ratio (c.f. (Steinley et al., 2012)) is applied on the dataset.
The linear projection does not reveal clear structures, even if the generalized U-matrix is applied to visualize high-
dimensional distance structures in the two-dimensional space. Therefore, it can be assumed that the structures cannot
be separated linearly, motivating the usage of more complex and elaborate methods. The clustering can be reproduced
with an accuracy of 86% using hierarchical clustering as described by Ward (Ward Jr, 1963) if the seven outliers are
disregarded because the Ward method is sensitive to outliers (Everitt et al., 2011). Silhouette plots (SI D, Fig. 11)
indicate inappropriate values for this clustering procedure if a spherical cluster structure is assumed. Statistical testing
indicates that the classes differ significantly from each other in the $NO_3$ and EC distributions (SI C), except for class 2
versus class 3 in NO3. However, class 2 and class 3 also differ significantly from each other in the variables of rain and
Wt18 (water temperature) in SI E, Fig. 12.

### 3.4 Step V: Knowledge Acquisition

The cluster structures are explained by applying the evtree (Grubinger et al., 2014) and CART algorithm (Breiman et
al., 1984; Therneau et al., 2018). The evtree decision tree is shown in Fig 7a and the CART decision tree for the data
is visualized in Fig. 7b. Both decision tree agree on the same features sets and relations for each cluster except for
cluster three for which Rain <0.2 is not required to differentiate from cluster one and two in evtree although that makes
cluster 3 less meaningful. The boundaries vary slightly between CART and evtree. None of the outliers could be
explained by either evtree or CART. CART has a lower error and improves the meaningfulness of cluster three.


Therefore, the rules are extracted from the CART tree instead of the evtree. The rules are used to describe the clusters

and could predict future $NO_3$ and EC values (Table 2). The description of class 2 gains more detail if maximum likelihood plots of rain and water temperature (Wt18) are used (SI E, Fig. 11). 3.5 Step VI: Knowledge Discovery

Next, we investigate the $NO_3$ and EC probability density distributions per class. In the last section, the clusters were explained by rules to define classes. The class-dependent MD-plots of Fig. 8 and Fig. 9 shown that the classes depend on normal or high $NO_3$ levels (Fig. 8) as well as on low, intermediate or high conductivity levels (Fig. 9) because the

distributions of classes differ significantly from one another, with the exception of $NO_3$ classes 2 and 3. This is confirmed by Kolmogorov–Smirnov tests (Supplementary C).

## 4 Discussion

Selecting a suitable distance measure enabled to apply the process of clustering of the above described dataset. The hypothesis in this cluster analysis was that intracluster distances are smaller (more similar to each other) than the

intercluster distances. As a parameter free clustering algorithm the DBS was chosen. It enables the evaluation of the clustering with a topographic map in addition to the conventional heatmap. DBS is a flexible and robust clustering framework that has the ability to separate complex distance-based structures [Thrun, 2018]. DBS consists of three interchangeable modules that are parameter free in projection and visualization (see Fig. 1). The second module is a human-understandable visualization technique (topographic map) that verifies the clustering/absence of clusters

(Thrun, 2018). Moreover, the number of clusters can be estimated prior to the clustering by the visualization of high-dimensional cluster structures. Such clusters structures were identified by low intracluster distances and high intercluster distances because the heatmap (Fig. 5) and topographic map (Fig. 4) showed clear cluster structures.

The heatmap and topographic map showed a valid clustering. However, simpler linear model (Fig. 6) or spherical cluster structure were inappropriate and models that assume an ellipsoidal structure would be insufficient (SI D, Fig.10,

SI B, Table 3). It follows, that most conventional approaches listed in (Aghabozorgi et al., 2015) would be not appropriate to detect data structures in the way we do here..

Statistical testing indicates that the distributions of relevant variables differ between classes (SI C and E). Further results imply that the clustering was meaningful because knowledge was extracted from the clustering by applying a classification and regression tree (CART) (Breiman et al., 1984) and using maximum likelihood plots. Overall, it can

be deduced that this dataset contains linearly non separable distance-based on non-spherical cluster structures. The acquired rules (Tab. 2) can be explained as follows:

While water temperature governs the biological turnover of nitrogen compounds in the stream water, hydrological variables such a groundwater level determine how and whether terrestrial $NO_3$ pools are connected to the stream system by activating flow pathways. Furthermore, the rainfall-runoff generation processes either concentrate or dilute the

stream $NO_3$ concentration, according to the difference in $NO_3$ concentration in the stream and in the "new water" added to the stream system.

In the search for days with similar behavior, days with normal and high $NO_3$ were identified. In 321 out of 343 days, the $NO_3$ concentrations were normal (in the average range of [1, 3.5] mg/L). On such days, the concentrations of electric conductivity (EC) were either high (in the average range of [0.034, 0.055] mS/m) or intermediate to low (in the average

range of [0.25, 0.045] mS/m). Normal $NO_3$ and higher EC occurred on dry days with increased stream water





temperature and higher groundwater levels. From a data-driven perspective, these days were highly similar to one another (c.f. cluster 1 in Fig. 4 and Fig. 4). The explanation for normal $NO_3$ with low to normal EC concentrations is more complex and described by "duality": they likely had an intermediate stream water temperature (6.1°C<WT18<12.5°C) with either dry days (average rain < 0.15 mm) and low groundwater levels (<1.28 m) or rainy

days with high groundwater levels (see SI E).

Simultaneously, high $NO_3$ concentrations (in the average range of [3, 5.5] mg/L) and very low EC concentrations (in the average range of [0.025, 0.028] mS/m) occurred only if the stream water temperature was low on dry days. In particular, stream water temperature influences the activities of living organisms. The groundwater level (or head, in m) is the primary factor driving discharge in the Schwingbach catchment, while rainfall intensity triggers discharge

and affects the leaching of nutrients (Orlowski et al., 2014).

**5 Conclusion**

No prior knowledge usable for a cluster analysis was available. Therefore a machine learned AI system, the Databionic swarm (DBS) method, was used for projection and clustering of environmental and water quality data. Rules extraction from a decision tree were applied on the clustering in order to explain the content of the clusters in a human

understandable way. The explanations suggest that the stream water quality data regarding $NO_3$ and EC can be described by a combination of one variable related to biological processes (water temperature) and two variables related to hydrological processes (rain and groundwater level). The understanding of these cluster structures and the application of the rules showed clear ranges of values and could enable future prediction of stream water quality.

The method presented here allows for unbiased detection of meaningful data structures in high dimensionality datasets.

Such datasets become more and more available, not only in hydrochemistry, but also in other environmental disciplines due to the technical innovation in monitoring equipment. Explainable AIs provide unique possibilities to search for unknown structures, but only if they do not rely on prior knowledge about data structure.

**Supplementary Information A: Features after Preprocessing**

Variables were preprocessed such that metric distances can be used because the range of every feature is approximately

between zero and one. The distribution of the features is shown by the MD-plot (Thrun et al., 2020) in Fig. 10 of the 12 variables used for DBS projection, visualization and clustering (Thrun, 2018). The complete aggregated dataset consisted of 343 days**.** In Fig. 8, the distribution of distances is visualized using the R package 'DataVisualizations' on CRAN (Thrun and Ultsch).

**Supplementary Information B: Comparison to Ward clustering approach**

The clustering can be reproduced with an accuracy of 86% using the Ward algorithm (Ward Jr, 1963) if the outliers are disregarded. The contingency table is presented below (Table 3).

**Supplementary Information C: Kolmogorov-Smirnov tests of clusters**

Table 4 and 5 compare the clustering achieved for conductivity and $NO_3$. The clusters should contain samples of different natures and based on different processes. Given this assumption, it is valid to statistically test whether the $NO_3$



and EC distributions significantly differ between clusters. The Kolmogorov-Smirnov test (KS test) is a nonparametric two-sample test of the null hypothesis that two variables are drawn from the same continuous distribution (Conover, 1971). For the first three clusters, the $NO_3$ and EC distributions significantly differ among clusters.

**Supplementary Information D: Silhouette Plot**

The Silhouette plot of DBC clustering is presented in Fig.11 and demonstrates an inappropriate clustering w.r.t.

spherical cluster structures.

**Supplementary Information E: Distinction of Classes 1 and 2 in Regard to Rain and Water Temperature**

Using the Kolmogorov-Smirnov test (KS test), which is a nonparametric two-sample test of the null hypothesis that two variables are drawn from the same continuous distribution (Conover, 1971), Class 1 significantly differs from Class 2 in the variable Wt18 (water temperature) with $p<(162,159, D= 0.31982)<0.001$ and in the variable rain with

$p<(162,159, D= 0.70498)<0.001$. This is visualized in the class wise maximum-likelihood plots of Fig. 12. Moreover, Fig. 12 (right) shows that the water temperature in Class 2 is more likely to be lower than that in Class 1 and less likely to be lower than that in Class 3.

**Code availability**

Every function used in this Manuscript is available in R packages on CRAN and is referenced throughout the text. The

specific application of these functions to the analyzed data is available in https://github.com/Mthrun/ExplainableAI4KnowledgeAcquisitionStreamTS2020/08AnalyseProgramme/. If not stated otherwise, no setting of parameters or changing of default parameters is necessary to reproduce the results above with the limitation that stochastic algorithms like most clustering and projection methods have a variance of results depending on the trial (c.f. discussion in (Thrun and Ultsch, 2020a)). The exact version of the model used to produce

the results used in this paper is archived on Zenodo: DOI: 10.5281/zenodo.3734892 under GPL license.

**Data availability**

The raw data is available on GitHub: https://github.com/Mthrun/ExplainableAI4KnowledgeAcquisitionStreamTS2020/90RawData/. Aggregated data is available in https://github.com/Mthrun/ExplainableAI4KnowledgeAcquisitionStreamTS2020/09Originale/. The exact

version of the model used to produce the results used in this paper is archived on Zenodo: DOI: 10.5281/zenodo.3734892 under GPL license.

**Author contribution**

Lutz Breuer devised the project and collected the data. Michael Thrun wrote the manuscript with major support from Lutz Breuer regarding all non-data science aspects. Michael Thrun designed the model and the computational

framework and analyzed the data. Alfred Ultsch supervised the project, checked calculations and programs and contributed and revised the manuscript. All authors discussed the results and contributed to the final manuscript.



**Competing interests**

The authors declare that they have no conflict of interest.

**Acknowledgements**

We thank Alice Aubert for fruitful discussions regarding the interpretation of the results.

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





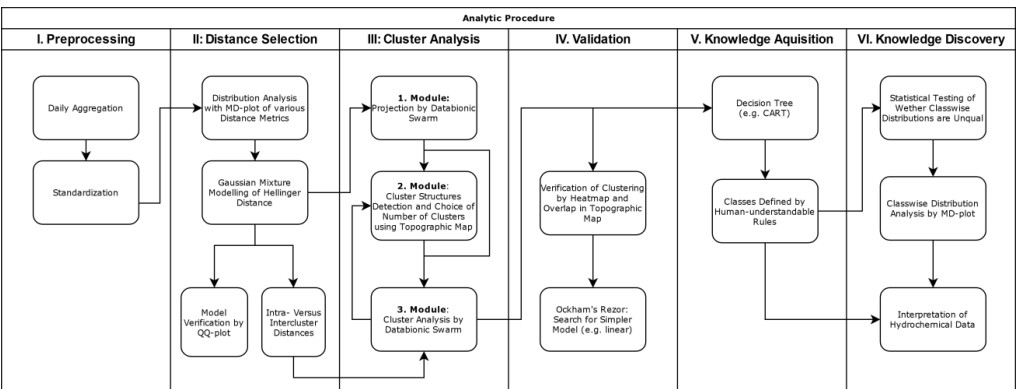

**Figure 1: Explainable AI for knowledge acquisition in stream time series without implicit assumptions about data structures (data-driven). Databionic Swarm clustering consists of three modules: the projection, inspection of cluster structures by the topographic map and the clustering.**

Table 1: Measured environmental variables with abbreviations and units. The probability density distributions of the

distributions of the transformed dataset are visualized in the supplementary section.

| Variable | Abbreviation | SI Unit |
|---|---|---|
| Soil temperature | St24 | °C |
| Groundwater level | GWl3 | m |
| | GWl25 | |
| | GWl32 | |
| Soil moisture | Smoist24 | m³/m³ |
| Rainfall | rain | mm/d |
| Discharge | q13 | L/s |
| | q18 | |
| Electric conductivity (EC) | Con47 | mS/m |
| Solar radiation | Sol71 | W/m$^2$ |
| Air temperature | At47 | °C |
| Streamwater temperature | Wt18 | °C |
| | Wt13 | |
| **Nitrate** (NO$_3$) | nnit13 | mg/L |

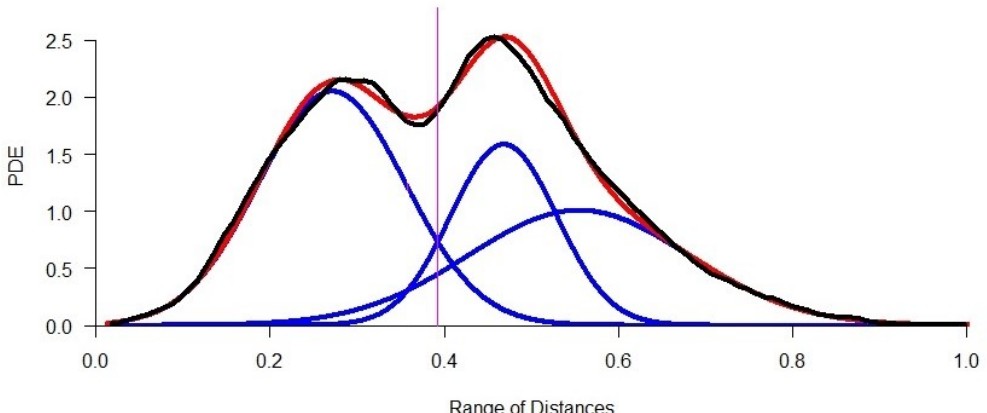

**Figure 2: Distribution analysis of the distances using a Gaussian mixture model (GMM) using the R package 'AdaptGauss'**
**available on CRAN (Ultsch et al., 2015). The first mode represents the intracluster distances, the second mode represents the intercluster distances, and the third mode indicates large outliers. PDE Pareto Density Estimation (Ultsch, 2005).**

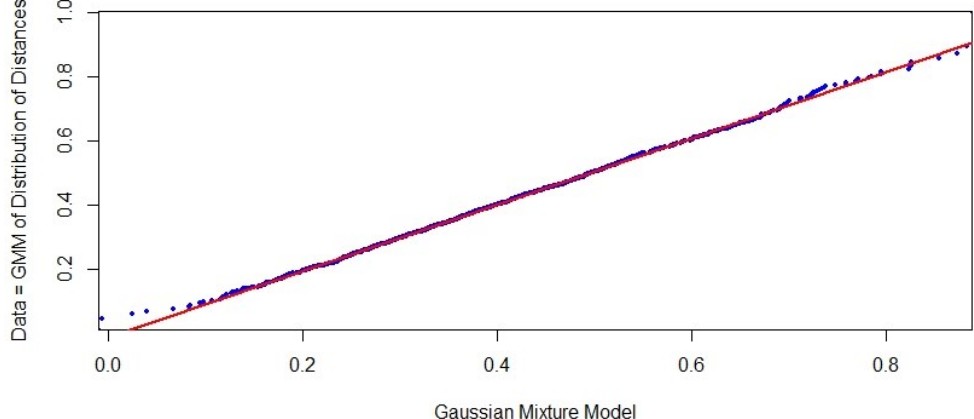

**Figure 3: Quantile-Quantile plot (QQ plot) visualizes a good match between the distance and the GMM through a straight line. Plot is generated using the R package 'AdaptGauss' available on CRAN (Ultsch et al., 2015).**

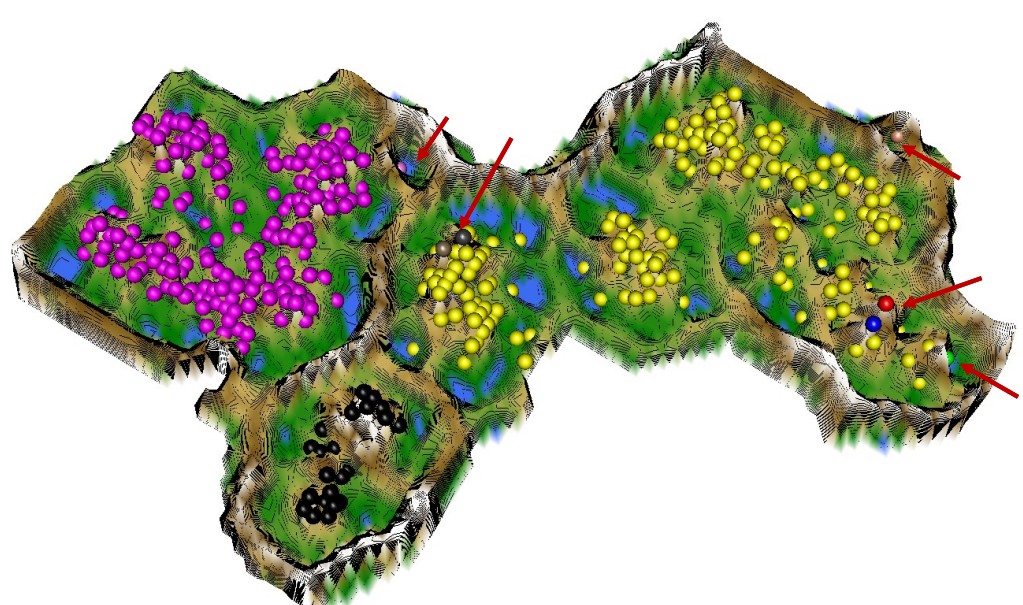


**Figure 4: The topographic map of high-dimensional structures shows two main clusters (magenta and yellow points), an outlier cluster (black points), and seven single outliers (marked by red arrows) in the hydrology dataset using the DBS method. Every point symbolizes a day, and the color of a point labels the cluster. Visualization of high-dimensional data structures is generated using the R package 'DatabionicSwarm' available on CRAN (Thrun, 2018).**

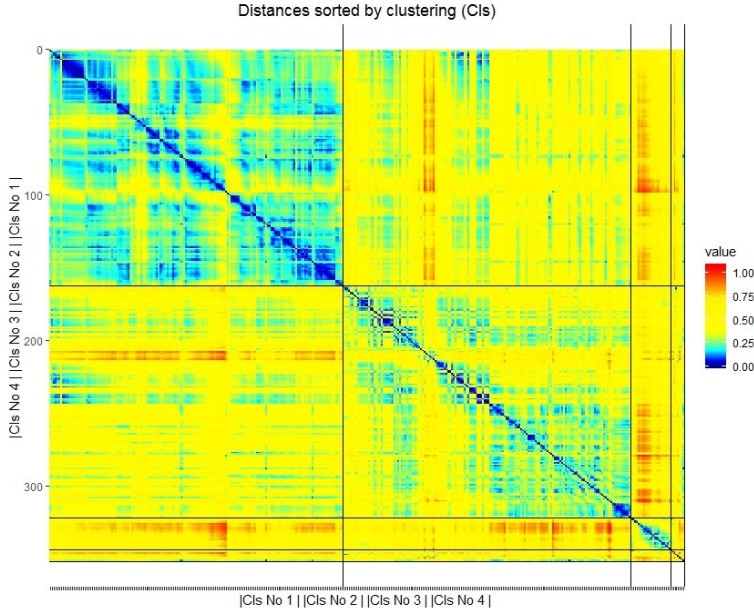


**Figure 5: The four clusters have distinctive distances, as shown by the heatmap. There are small distances within each cluster and large distances between the clusters. The outliers are summarized in Cls4. The heatmap was generated with the R package 'DataVisualizations' available on CRAN (Thrun and Ultsch, 2018).**

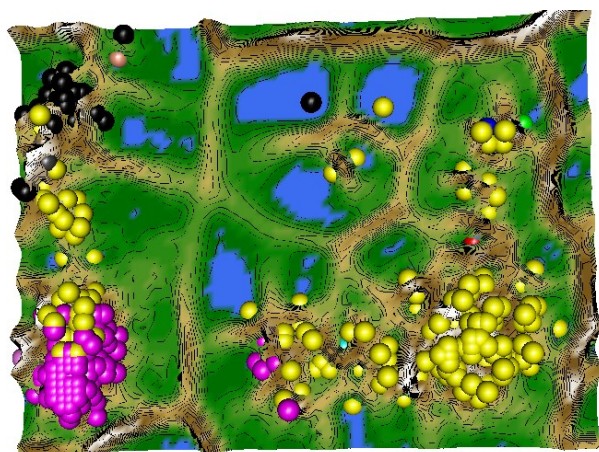

Figure 6: Toroidal topographic map of a projection pursuit approach by (Hofmeyr and Pavlidis, 2015) of the hydrology dataset. The linear projection does not reveal a linear structure, even if the generalized U-matrix is used to visualize high-dimensional distances of the two-dimensional projection (Ultsch and Thrun, 2017).

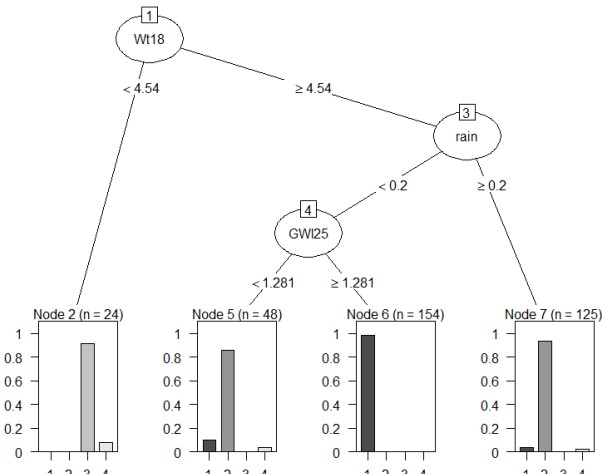

Figure 7a:        Globally optimal classification and regression trees (evtree) analysis rules for the dataset with the three clusters identified by the DBS method. The error of class 1 is 15%, class2 is 6.4% and class 3 is 8.3%. Outliers are summarized in class 4. The rules are quite similar to Fig 7B but have a higher error.

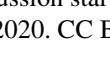



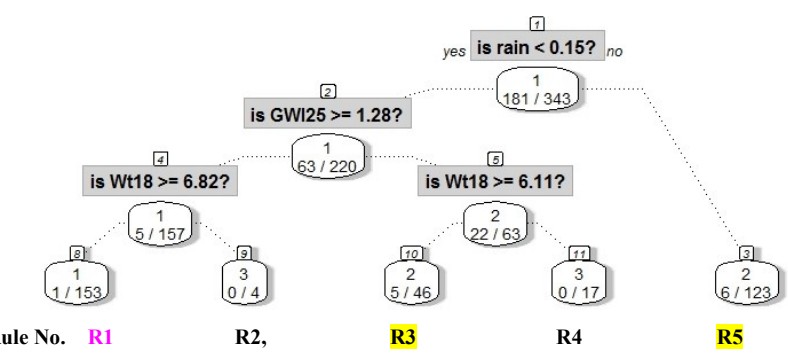

**Figure 7b:** Classification and regression tree (CART) analysis rules for the dataset with the three clusters identified by the DBS method. Applying the rules to the clustering in combination with the dataset results in 12 misclassified points (3.5% of daily observations). 8 outlier points are in class 4 for which nodes can be derived. This error is lower than in Fig7a. For units of measurements and abbreviations, please see table 1.

**Table 2: The CART rules based on Fig. 7. Abbreviations: rainfall intensity (rain), soil temperature (St24), soil moisture (Smoist24), and water level at point 3 (GWl3). All values are expressed as percentages. For units of measurement, please see table 1. Class 2 R5 is extended by SI E, Fig. 11.**

| Rule No. | DBS Cluster No. | No. of Days | Rule | Short Abbreviation for Subsequent Plots |
|---|---|---|---|---|
| R1 | 1 | 162 | rain < 0.15 and GWl25≥1.28 and Wt18 ≥6.86 => Dry days, increased stream water temperature and groundwater levels | *DryDaysWarmWater* |
| R3 & R5, Fig. 11 | 2 | 159 | rain < 0.15 and GWl25 < 1.28 and Wt18 ≥6.11 or rain >= 0.15 and Wt18 ≥6.11 => Intermediate stream water temperature with either dry days and low groundwater levels or rainy days with a high level of water | Duality |
| R2 & R4 | 3 | 22 | rain < 0.15 and GWl25 ≥ 1.28 and Wt18 < 6.86 or rain < 0.15 and GWl25 < 1.28 and Wt18 < 6.11 or => Dry days with colder stream water and variable groundwater levels | *DryDaysColdWater* |
| - | Unclassified | 7 | excluded because cannot be explained with CART | Outliers |

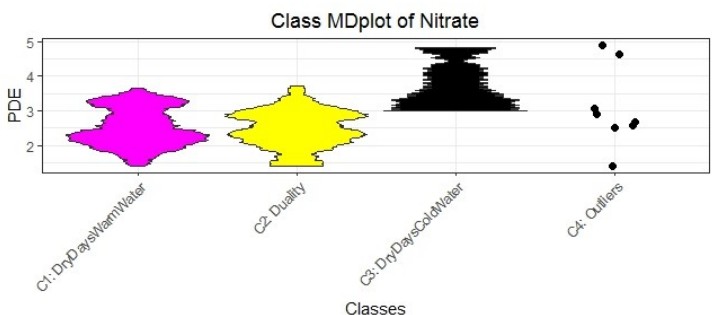

**Figure 8:** Class wise mirrored-density plot (MD-plot) of the three classes with regard to $NO_3$ and the outliers. There are two low to intermediate classes of N concentrations and one class of high N concentrations. Classes are colored similar to the clusters in Fig. 4. The MD-plot was generated using the R package 'DataVisualizations' available on CRAN (Thrun et al., 2020).

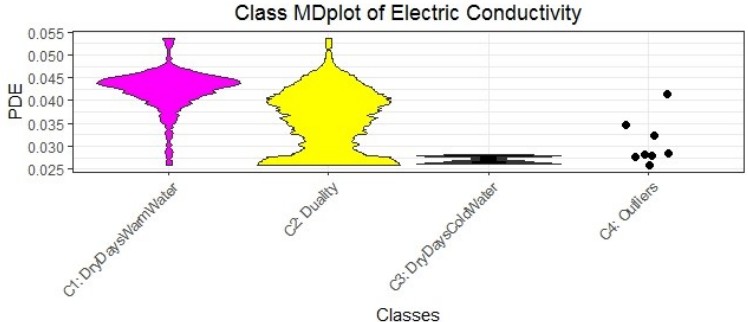

**Figure 9:** Class wise mirrored-density plots (MD-plot) of the four classes with regard to electrical conductivity C. There is a class of high concentration, a class of low to intermediate concentration and a class of very low C concentrations. Classes are colored similar to the clusters in Fig. 4. The MD plot was generated using the R package 'DataVisualizations' available on CRAN (Thrun et al., 2020).



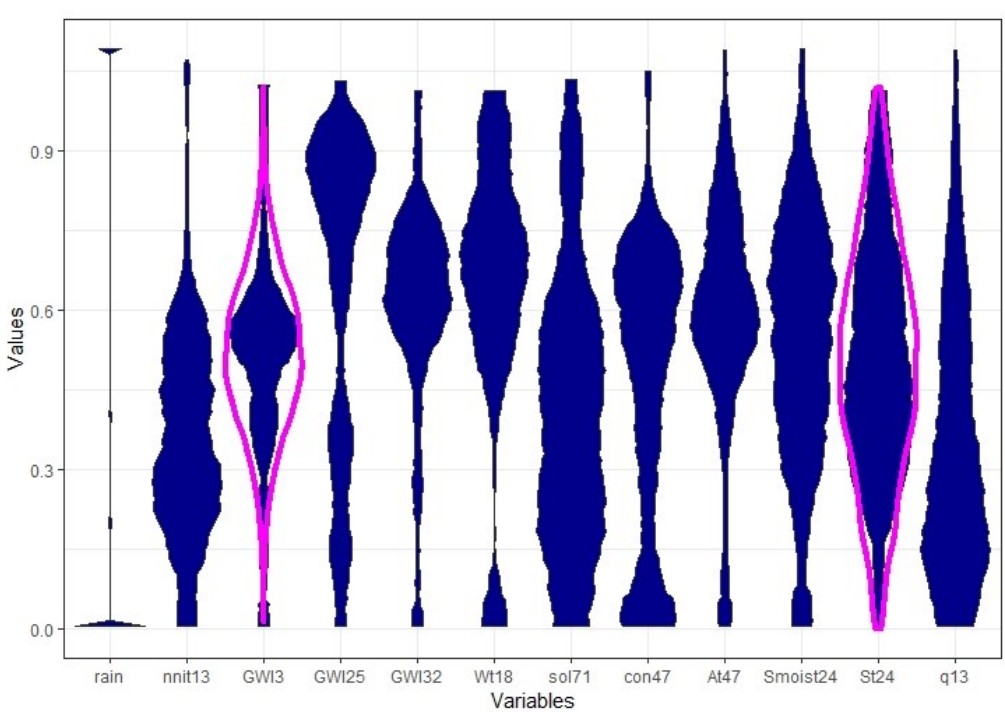

**Figure 10: The distribution of variables after preprocessing is visualized using mirrored-density plots of the hydrology dataset. The magenta overlay marks features that are statistically not skewed or multimodal. The mirrored-density plot (MD-plot) was generated using the R package 'DataVisualizations' available on CRAN (Thrun and Ultsch, 2018).**


**Table 3: Contingency table of DBS versus Ward.**

| DBS/Ward | 1 | 2 | 3 | RowSum | RowPercentage |
|---|---|---|---|---|---|
| 1 | 157 | 5 | 0 | 162 | 47.23 |
| 2 | 42 | 116 | 1 | 159 | 46.36 |
| 3 | 0 | 0 | 22 | 22 | 6.41 |
| ColumnSum | 199 | 121 | 23 | 343 | 0 |
| ColPercentage | 58.02 | 35.28 | 6.71 | 0 | 100 |


**Table 4: KS test with test statistic ($D$) and p-value ($p$) for conductivity for the first three clusters.**

| Cluster No. (Sample Size) | C2 (159) | C3 (22) |
|---|---|---|
| C1 (162) | $D=0.13429$, $p=0.11$ | $D=0.74074$, $p<0.001$ |
| C2 (159) | | $D=0.84906$, $p<0.001$ |





**Table 5: KS test with test statistic (*D*) and p-value (*p*) for NO₃ for the first three clusters.**

| Cluster No. (Sample Size) | C2 (159) | C3 (22) |
|---|---|---|
| C1 (162) | *D=0.50769, p<0.001* | *D=0.98765, p<0.001* |
| C2 (159) | | *D=0.83019, p<0.001* |

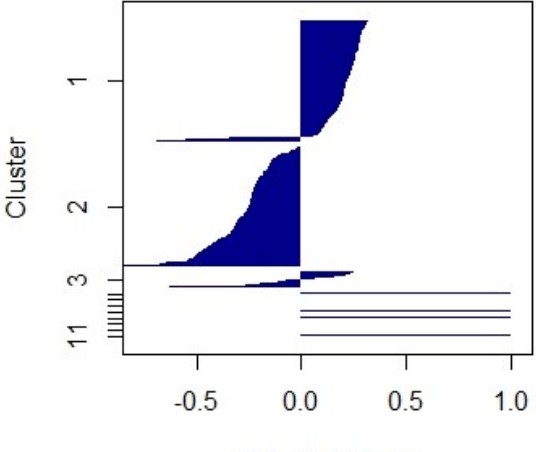


**Figure 11: Silhouette plot of DBC clustering shows low values for the three main clusters, indicating inappropriate clustering with regard to expected spherical structures. The silhouette plot was generated using the R package 'DataVisualizations' available on CRAN (Thrun and Ultsch, 2018).**

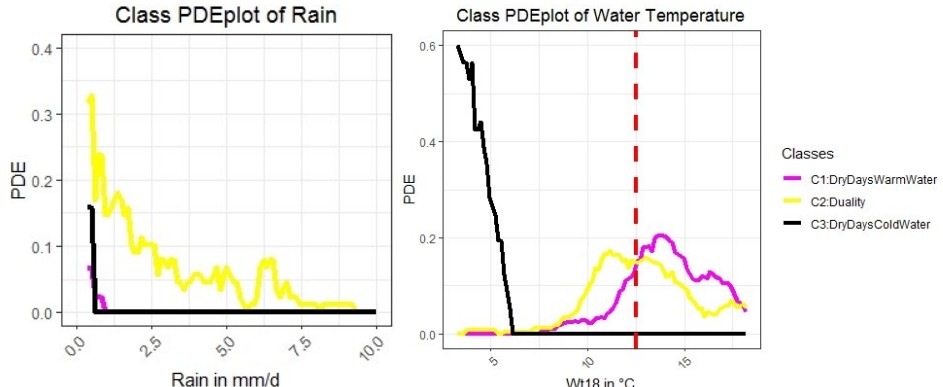


**Figure 12: Class wise estimation of the probability density function using PDE allows for a more precise definition of Class 2, "Duality", because the plot shows that in Class 2, there are also rainy days with colder water than in Class 3. The red and dashed line in the right plot marks a temperature of 12.5°C. Classes are colored similarly to the clusters in Fig. 4.**