# Peer review of "Explainable AI for Knowledge Acquisition in Hydrochemical Time Series V1.0.0"

_Geoscientific Model Development, 2020_

## Referee Comment (RC1) · Anonymous Referee #1 · 14 Jun 2020

General Comments:

Overall, I would say that this paper has an interesting concept but is hard to interpret the main contributions and is overall hard to read. For example, in Section 2 there needs to be more information about Figure 1 in this section (Material and Methods), the authors go straight into the steps (1-6) without a high-overview first. I was just confused of the paper structure without reading Fig. 1 first, which is hard to interpret without a better explanation. There are many arrows and connections that are not explained well, because of this, I don't really have a good understanding the main concepts and think the authors need to make things clearer and easier to interpret. I just would like to have a better understanding of how decision trees can be used for DBS and other methods (e.g. k-means) and believe if the authors can make this clearer then it would

improve the overall paper. For example, can you add a Figure/Algorithm that explains Step 3 to Steps 5/6 better?

Other Comments:

This lacks recent explainable frameworks for clustering, for example the paper "An Explainable Artificial Intelligence Model for Clustering Numerical Databases" is very similar and does a nice job at comparing other clustering approaches (k-means) too.

I also think the authors should either explain better why this is considered an AI system. I don't really understand this until Section 2.3 (cluster analysis) where they mention that DBS uses a mix of game-theory and neural networks. This information should made more clear in the Introduction.

Section 2.1: Where are this values collected at (Germany?)? There should be a figure of the location and points on a map, this would help with the interpretation of the topographic results. Do you need to state the variables that were removed from the analysis since they have high-overlap of similarity? Shouldn't the AI system do this for you?

Section 2.2: Why is a distance metric important for the algorithm, this isn't explained well. For example, I think you need to explain the DBS system (Section 3) before this section stating why this important for the algorithm.

Figure 1: This needs to be made clearer. This is hard to interpret and not sure what's going on.

Figure 2: This needs a legend.

Figure 4: How many clusters are here? There needs to be a legend.

Figure 5: This is hard to interpret. Where are the four clusters? Can you add these to the heat map?

Figure 6: Do you need this? Seems not really relevant to the paper or combine this

with Figure 4.

Figures 8/9: Seem like these can be merged (Fig 8 a/b).

Figure 10: Is this important? Seems like this can be excluded.

Labeling of the Figures are off, Figure 6 is mentioned in the Discussion after Figure 9.

Discussion: Is it possible to include other methods (k-means, DBSCAN) in this framework?

---

## Referee Comment (RC2) · Anonymous Referee #2 · 1 Jul 2020

General Comments: This manuscript takes watershed time series data for nitrate concentration (NO3), electrical conductance (EC), and 12 other typical hydrology-related parameters from a previously published data set (reported on by Aubert et al., 2016) and attempts to understand conditions which lead to high and low nitrate and EC levels through the application of an AI procedure. The concept is straightforward and the value of deriving insight from large data sets is critical. The current manuscript suffers from poor description of methods and processes and could be improved through re-organization and removal of notable errors in grammar and writing. In order to follow the paper, I frequently had to go back and forth between sections and had to read the entire previous publication about the data themselves (Aubrey et al., 2016) in order to evaluate the current manuscript. What is perhaps what I find most troubling with the

current work is that the key findings of the study are not new and are concepts can be found in undergraduate level courses covering on water quality in natural systems.

Major/Global Comments:

1) The description of the data used in the study is poor. From the introduction, it is stated that the data come from the Swingbach catchment in Germany, and from the methods section that there are 32,196 data points for 14 variables. No discussion about how many sites there, where the data were collected, or if the data even overlap temporally. Instead of using standard terms for things like groundwater level or stream temperature, they are replaced with cryptic codes (e.g., groundwater level = GW13, GW125, or GW 132) that are make it difficult to follow the results. It took reading the previous paper to produce even a moderate understanding of what the variables are and why there appear to be duplicates. In terms of understanding underlying hydrological process, being able to discern that, for instance, that the three groundwater level measurements are for a lowland, hill slope, and riparian zone better tremendously.

2) Despite having a reasonable background, in terms of both the data and the methods applied in this paper, the current text provides a poor explanation of work that is spread out throughout the text. Take for instance a very simple examination of Figure 2, which plots probability distribution estimators of "distance" for the data, as well as Gaussian end-member populations which fit the data. But what distance? There are 14 variables in the study, so is this a multivariate distance of some sort? The corresponding text simply states that, "The Hellinger distance measure is selected. . . " (line 93) but Hellinger distance measures distance between distributions. So which distributions is it measuring distances between? Going back to the paper by Auger et al. (2106), they observed a tri-modal distribution of NO3 concentrations. . .is that what this is?

In another example, we can examine how the cluster analysis is explained (note that the section title is misspelled as "Clsuter Analysis). It begins by focusing on the Pswarm method, where DataBots move data that are similar towards one another on a grid or

map, but this is described using only qualitative means, such as searching for the "most potent scent" or moving towards DataBots with the most similar features. What type of similarity metric(s) is employed here? Later in the same discussion, it is explained that clustering can either be focused on compact clusters or connected clusters and that the authors have decided to emphasize the former (no justification given). The following text (line 14) then goes on to describe how "the choice of this parameter can be evaluated...". What parameter? Is this related to compact clusters or connected clusters?

The final issue about the methods is that the reader basically must go back and forth through the paper to follow what was done and the arguments behind it. In the methods section on data pre-processing, mirrored density plots are described as being employed but no details about what they are or why they are used is provided (they look similar to violin plots but turn out to be somewhat different). However, a fuller explanation is provided much later, in the final paragraph of the methods, instead of when they are first introduced.

3) This work relies on and cites a large number of packages in R. While there's nothing fundamentally wrong with that, citing a package without describing the methods it relies on is not beneficial. My recommendation is that a new table be created which lists all the packages used and their citations. Then in text, simply state the package title and what principles or techniques the package uses.

4) The fundamental processes identified by this investigation are largely rote conclusions for scientists who study water quality in paired stream-aquifer systems. For instance, input of groundwater into a stream corresponds generally with higher temperature (due to geothermal contribution) and EC (due to longer residence time to allow for water-rock interaction). Similarly, high nitrate levels during dry days and lower stream temperature is due to a lack of dilution effect during rainfall, bur still primarily a surface water vs. groundwater contribution to streams (i.e., lower temperature). What is the value added by this analysis?

I have numerous specific comments and noted many typographical errors as well, but the feel that the items above need to be rectified before providing further feedback.

---

## Author Comment (AC1) · 31 Jul 2020

General Comments:

Overall, I would say that this paper has an interesting concept but is hard to interpret the main contributions and is overall hard to read. For example, in Section 2 there needs to be more information about Figure 1 in this section (Material and Methods), the authors go straight into the steps (1-6) without a high-overview first. I was just confused of the paper structure without reading Fig. 1 first, which is hard to interpret without a better explanation. There are many arrows and connections that are not explained well, because of this, I don't really have a good understanding the main concepts and think the authors need to make things clearer and easier to interpret.

Dear Reviewer #1,

Thank you for the very valid feedback that will allow us to improve our manuscript significantly. In a revised manuscript, we will provide more information on Figure 1 and include the following text:

"Each step progresses from top to bottom. Arrows outline the connections between the steps. In the first step, the time series data is aggregated appropriately (e.g., daily) and then standardized. In step II, various available distance metrics are applied to the data and investigated for multimodality. If a distance distribution is multimodal, it can be modeled by a Gaussian Mixture. This distance should be prefered for the cluster analysis in the third step. If not distance is multimodal, the framework continuous with the Euclidean distance. The cluster analysis in step III is composed of three modules. It starts with the projection in module 1and follows the structure visualization through a topographic map in module 2. This visualization enables the user to choose the number of clusters and the setting of the Boolean parameter. One arrow form module 1 bypasses module 2 and points directly to module 3, meaning that cluster analysis can be performed independently of the visualization. From module 3, an arrow points back to module 2, indicating thatthe number of clusters can be set as the number of visible valleys. The clustering of module 3 can be further evaluated by the model of the distance distribution (last arrow between steps II and III), because using the Gaussian mixture model of step II hypothesizes that the intracluster distances of the clustering should be smaller than the Bayesian boundary defined by the Gaussian Mixture. Module 3 can be changed to another clustering algorithm accordingly to the preference of the user. In step IV the clustering can be validated by the topographic map. Additionally it is preferable to search for linear models and to validate the clustering externally (e.g. Heatmap). After validation in step IV, the resulting clustering of step III is used in a supervised Classification and Regression Trees (CART) analysis for the training of the un-preprocessed but aggregated data in step V. Then, the classes are defined by rules which are defined by paths in the decision tree. In the last step VI, the classes and their rules are interpreted by domain experts (c.f. [Miller, 2019]). Class-wise distribution analysis and statistical testing of relevant features can be performed to assure that explanations are tendentially contrastive (c.f. [Miller, 2019]). The details of the analytic procedure can be found in the methods sections, which is organized according to these six steps, as illustrated in the titles of the various steps in Fig. 1."

I just would like to have a better understanding of how decision trees can be used for DBS and other methods (e.g. k-means) and believe if the authors can make this clearer then it would improve the overall paper.

We will include the following statement on how decision trees are included in the DBS:

"The conventional usage of decision trees is either supervised, requiring a prior classification (e.g. [Breiman, 2001]) or unsupervised using split evaluation criteria that does not require a prior classification

(e.g. [Basak/Krishnapuram, 2005]). Here, we propose a third approach by using a cluster analysis in step III instead of a prior classification. Contrary to common usage, the decision tree is exploited here to explain the clusters by transforming decision tree paths to rules. The rules describing the clusters allows the user finally to define classes."

For example, can you add a Figure/Algorithm that explains Step 3 to Steps 5/6 better?

We hope that the general description of Figure 1 is now clearer, particularly by elaborating steps III to VI. It should be noted that Figure 1 had a minor mistake (i.e., an arrow from module 2 to module 3 in step III, which will be deleted in the revision).

Other Comments:
This lacks recent explainable frameworks for clustering, for example the paper "An Explainable Artificial Intelligence Model for Clustering Numerical Databases" is very similar and does a nice job at comparing other clustering approaches (k-means) too.

The reviewer makes a very valid point. We would like to add to the manuscript a related work subsection presenting existing explainable frameworks, focusing on decision trees:
 "There are two approaches for the explanation of machine learning systems: prediction, interpretation and justification that is used for subsymbolic ML systems (defined in [Alfred Ultsch, 1998]) and interpretable approaches for symbolic ML systems (defined in [Alfred Ultsch/Korus, 1995]), which are explained through reasoning [Biran/Cotton, 2017]. For the former, a well-known example is LIME [Ribeiro et al., 2016], which approximates any classifier or regressor locally with an interpretable model. In the sense of the latter, explanation represents a distinct approach to extract information from the learned model [Lipton, 2018]. Typical interpretable ML systems comprise combinations of neural networks with rule-based expert systems [A Ultsch et al., 1991; Alfred Ultsch et al., 1995], Bayesian networks with rule mining [Letham et al., 2013], hybrids of clustering and fuzzy classification [Riid/Sarv, 2013] or neuro-fuzzy classification [Nauck/Kruse, 1999], interpretable decision sets [Lakkaraju et al., 2016] or decision tables [Hewett/Leuchner, 2002], decision tree clustering [Basak/Krishnapuram, 2005] or clustering combined with generative models [Kim et al., 2015].
One of the most recent approaches, decision tree clustering UD3.5, is a hybrid of k-means and a top-down decision tree with a split evaluation criterion inspired on the silhouette plot because decision trees can be modified by using supervised quality measures [Loyola-González et al., 2020]. These authors claim to have similar performance to k-means and better performance than other conventional decision tree clustering algorithms [Loyola-González et al., 2020]. The major difference to the approach follwed here is that Loyola-Gonzales et al. use unsupervised decision trees whereas this work uses supervised decision trees based on a clustering that is performed independently. This has several advantages. For example, it was shown in Thrun and Ultsch, that k-means only is able to grasp spherical cluster structures, which is a severe restriction [Thrun/Ultsch, 2020b]. Moreover, the Silhouette plot is only useful in the case of spherical or ellipsoidal structures [Rousseeuw, 1987; Herrmann, 2011], leading to the assumption that UD3.5 will prefer hyper ellipsoidal cluster structures. In contrast, the Databionic Swarm presented here outperforms k-means because it can find a large variety of cluster structures through the exploitation of emergence and self-organization [Thrun/Ultsch, 2020a]. These authors state that "Additionally, [clustering algorithms] may return meaningless results in the absence of natural clusters [Cormack, 1971, pp. 345-346; Jain/Dubes, 1988, p. 75; Handl et al., 2005, p. 3203] in contrast to DBS which will clearly indicate that no cluster structures exist" [Thrun/Ultsch, 2020a]. Moreover, DBS is able to discover small classes [Thrun/Ultsch, 2020a] whereas UD3.5 is not [Loyola-González et al., 2020] (p. 52381). Finally, DBS is open

source which is not the case for UD3.5.

Furthermore, the authors of this work agree with Mörchen and Ultsch, that explainability should follow the Gricean maxims of quality, relevance, manner, and quantity [Mörchen/Ultsch, 2007]. The maxim of quality states that only well-supported facts and no false descriptions should be reported. The maxim of relevance requires that only rules relevant to the expert are listed. The maxim of manner suggests to be brief and orderly and to avoid obscurity and ambiguity. The maxim of quantity states that neither too much nor too few rules should be presented [Mörchen/Ultsch, 2007]. In comparison to [Loyola-González et al., 2020], explainable rules upon Gricean maxims are presented in [Basak/Krishnapuram, 2005](table 3 in their work) . In our work the rules are brief and relevant whereas for explainable UD3.5 (Table 3 in their work) the rules are not brief and relevant because many rules are presented for each cluster which do not cover the clusters well enough. "

I also think the authors should either explain better why this is considered an AI system. I don't really understand this until Section 2.3 (cluster analysis) where they mention that DBS uses a mix of game-theory and neural networks. This information should made more clear in the Introduction.

We agree and will include a statement on the specific features that describes our interpretable machine learing system. This will read:

"The framework based on DBS presented here includes a mix of game-theory, neural networks and supervised decision trees. These are combined in a sophisticated manner to detect so far unknown relationsships in data which makes it a comprehensive new tool for interpretable machine learning or so called explainable AI systems. It should be noted that interpretable ML systems or explainable AIs (XAIs) are used synonymously [Adadi/Berrada, 2018]."

Section 2.1: Where are this values collected at (Germany?)?
There should be a figure of the location and points on a map, this would help with the interpretation of the topographic results.

We will include a new section in the Material and Methods chapter on data collection, further technical information on analytical approaches as well as a site description:

"Data used in this work have been collected in in 2013/2014 in the Schwingbach Environmental Observatory (SEO), in central Germany. The mixed developed landscape is mainly characterized by agricultural land use (44%) and forests (48%). An in-situ hyperspectral UV-spectrometer (ProPS, Trios, Rastede, Germany, wavelength range 200–360 nm, path length 5 mm, solar panel supplied) was used to measure absorption spectra every 15 min. Prior to measurements, air blasts (5 s) were blown on the lens to prevent the optics from biofouling. Wavelengths spectra at 200–220 nm were utilized for the calculation of nitrate concentration, following a calibration with local stream water matrix (see Aubert et al. 2016 for further details). All other variables used in this machine learning approach (Table 1) were also monitored at the same high-frequency or aggregated to 15 min intervals. Discharge and stream water temperature were recorded by pressure transducers (Diver DCX, Schlumberger Water Services, ON, Canada) at two gauging stations at the outlet (q13) and upstream (q18) of the first-order stream of the Vollnkirchener Bach. Groundwater depth at three wells (GW25 hillslope, GW3 lowland, GW32 riparian zone) were measured with similar pressure transducers. Electromagnetic induction sensor (5TE) attached to EM50 data loggers (Decagon, Labcell LTD, Alton, UK) installed at 0.1 m depth in the riparian zone were used to gauge soil moisture and soil temperature. All meteorological data was collected at a climate station 4 km from the outlet (Campbell Scientific Inc., CR1000 data logger, Loughborough, UK).

In total, the dataset contains 32,196 data points for 14 different variables. Data gaps due to technical

problems and data quality control reduced the available data coverage to two growing seasons (05 March 2013 to 24 September 2013, n=15,475 measurements; 27 April 2014 to 23 October 2014, n=16,721 measurements). In Table 1 abbreviations and SI units of all variables are provided. Further technical information on the SEO, the analytical procedures applied, the coding of abbreviations and the experimental design in general are outlined in detail by Aubert et al. 2016, 2017 and Orlowski et al. 2014."

Do you need to state the variables that were removed from the analysis since they have high-overlap of similarity? Shouldn't the AI system do this for you?

We are not sure if we understand the question of the reviewer correctly. If the question is in context of the preprocessing, then yes, we should state which variables are removed and why they are removed. The reason is that high correlating variables have to be accounted for because otherwise they are over-weighted in the following analysis. We will include a statement on this in the revised manuscript in the description of step I in chapter 2:
"Correalting variables have to be detected before further data evaluation as otherwise these variables will be over-weighted in the assessment of the following distance metrices."

Section 2.2: Why is a distance metric important for the algorithm, this isn't explained well. For example, I think you need to explain the DBS system (Section 3) before this section stating why this important for the algorithm.

We will add to section 2.2. the following paragraph to elaborate the importance of the distance metric:
"Usually, partitioning and hierarchical clustering algorithms require a distance metric because they seek to find groups of similar objects [Bouveyron et al., 2012] (i.e., objects with small distances between them). If no specific distance measure is used, most algorithms use the Euclidean distance metric, and the user is not always able to change the distance metric manually (c.f. 54 common algorithms in https://CRAN.R-project.org/package=FCPS). DBS has the advantage that a specific distance metric can be selected by the user. The distance is integrated into the scent around each agent called DataBot [Thrun/Ultsch, 2020a] (p. 18 Equation 5). However, the choice of distance remains undiscussed in prior work. We propose that a user selects a distance metric based on specific properties of the distance distribution of the specific data set."

Figure 1: This needs to be made clearer. This is hard to interpret and not sure what's going on.

Please refer to our first reply to Figure 1 above.

Figure 2: This needs a legend.

We will include a detailed explanation of the colored lines in the revised figure caption of Figure 2. Hence, an additional legend is not necessarily needed but relevant information can be integrated into the figure caption. The revised figure caption will read:
"Figure 2: Distribution analysis of the distances using a Gaussian mixture model (GMM) using the R package 'AdaptGauss' available on CRAN (Ultsch et al., 2015). Black line indicates the estimated distribution of the distance feature *df* (defined in SI F). The distribution is estimated using PDE. The blue line depicts the single Gaussian distributions ("modes") of the model and the red line the overall model,

i.e., the superposition of single Gaussians to a mixture. Bayes Boundary in magenta separates the first mode from the second mode and the third mode leading to the hypothesis that the first mode should consist of intra-cluster distances, if a clustering is performed. PDE=Pareto Density Estimation (Ultsch, 2005). "

Figure 4: How many clusters are here? There needs to be a legend.

In total, we found three clusters which was already mentioned in the previous figure caption. We do not think that a legend to this figure is needed, as the color of clusters and the heights of the valleys and tops of the topographic map only have a qualitative meaning as described in section 2.3. .

Nevertheless, we will add the following statement to section 3.2:
"Projection points near to each other are not necessarily near in the high-dimensional space (vice versa for far away points) but in planar projections of data these errors are unavoidable (c.f. Johnson-Lindenstrauss Lemma [Johnson/Lindenstrauss, 1984]). Hence, the topographic map of high-dimensional structures evalutes the clustering by indicating which points are in the high-dimensional space far away (brown/white hills) or near (blue seas, green grassland). "

We will further revise the figure caption to better explain this. It will read:
"Figure 4: Every point symbolizes a day and is colored by the indepently performed clustering. Clusters lie in valleys. Three clusters with two major clusters (magenta and yellow points) and one cluster of outliers (black points) are visible. In addition, seven single outliers (marked by red arrows) are depicted. Visualization of high-dimensional data structures is generated using the R package 'DatabionicSwarm' available on CRAN (Thrun, 2018)."

Figure 5: This is hard to interpret.

We will add in section 2.4. an extensive introduction to the cluster heatmap as follows:
Engle et al state: "Cluster heatmaps are commonly used in biology and related fields to reveal hierarchical clusters in data matrices. Heatmaps visualize a data matrix by drawing a rectangular grid corresponding to rows and columns in the matrix and coloring the cells by their values in the data matrix. In their most basic form, heatmaps have been used for over a century [Wilkinson/Friendly, 2012]. In addition to coloring cells, cluster heatmaps reorder the rows and/or columns of the matrix based on the results of hierarchical clustering. (…). Cluster heatmaps have high data density, allowing them to compact large amounts of information into a small space [Weinstein, 2008]."[Engle et al., 2017]. For distance matrices, the procedure can be performed as described above Meaning that distances are ordered by the clustering, each pixel represents an distance valuem, and the clusters are divided by black lines."

Where are the four clusters? Can you add these to the heat map?

We will revise Figure 5 accordingly and change the "Cls" shortcut to cluster 1, 2,3 and 4 (outliers). In order to be reproducible, we will revise the subsequent method DataVisualizations::Heatmap() leading to a clearer legend and an improved documentation. The revised Fig. 5 will be added in the revised version of the manuscript and is attached to this answer. In addition, we willd extend the description of step IV in

section 2.4 (see above) and add an explanation to section 3.3:
"The heatmap depicts the homogeneity of clusters because the pattern of blue and teal colors is present for intra-cluster distances and yellow to red color pattern for inter-cluster distances."

Figure 6: Do you need this? Seems not really relevant to the paper or combine this with Figure 4.

We agree that Figure 6 is not of utmost importance. Nevertheless, we would like to keep the information that simpler clusters are not necessarily better ones, which is in agreement with our statement on Ockham's razor. We therefore will move this Figure to the supplement.

Figures 8/9: Seem like these can be merged (Fig 8 a/b).

We will merge both figures as recommended.

Figure 10: Is this important? Seems like this can be excluded.

We stated in step I that data preprocessing is required, but did not explain in detail why it is important. We therefore will include the following description to section 2.1, which relates to the need of keeping Figure 10:
"Distance measures are sensitive to the variance in the distribution of features. For example, the Euclidean metric weights feature more if they have values above 1. Therefore, the variance of features is usually standardized before a cluster analysis is performed. Fig. 10 shows the result of an appropriate standardization of features resulting in similar variances. It should be noted that for the explanation of the clustering as described in step V, non-standardized features are preferable because the interpretable ML system should explain the clustering to the domain expert and not the data scientist [Miller et al., 2017]."

Labeling of the Figures are off, Figure 6 is mentioned in the Discussion after Figure 9.

In the results section, the reference to Fig. 6 is missing in Line 219 . When the corrected reference is added in the revised version, the labelling of the figures will not be off.

Discussion: Is it possible to include other methods (k-means, DBSCAN) in this framework?

Yes, further methods can be included. Examplarily, we applied a more simple clustering method based on a linear projection method of [Hofmeyr/Pavlidis, 2015] and presented results in in Fig, 6. We will add the following statement to the discussion:
"It should be noted that module 3 of the DBS shown in step III (see Figure 1) can be changed to another clustering algorithm such as k-means of DBSCAN [Thrun/Ultsch, 2020a], which most-often results in a focus on specific cluster structures [Thrun/Ultsch, 2020b]. This can be preferable if prior knowledge about the data exists that leads a user to a specific choice of a global cluster criterion."

**Anonymous Referee #2**

General Comments: This manuscript takes watershed time series data for nitrate concentration (NO3), electrical conductance (EC), and 12 other typical hydrology-related parameters from a previously published data set (reported on by Aubert et al., 2016) and attempts to understand conditions which lead to high and low nitrate and EC levels through the application of an AI procedure. The concept is straightforward and the value of deriving insight from large data sets is critical.

Dear Referee #2,
Thank you for your extensive work you put into understanding the algorithms we developed and applied in our work.

The current manuscript suffers from poor description of methods and processes and could be improved through re- organization and removal of notable errors in grammar and writing. In order to follow the paper, I frequently had to go back and forth between sections and had to read the entire previous publication about the data themselves (Aubrey et al., 2016) in order to evaluate the current manuscript.

We will overhaul the methods section in various places to improve is logical order. We will further overhaul Figure 1 and improve parts of the description of the work flow in the respective six steps of our method. We see these six steps as a guidance for a reader had to build an explainable AI (interpretable machine learning system) for her/his dataset. We will further include a detailed description in Section 2.1 on the monitoring program and field analytical procedures we used. Finally, we acquired the service of Springer nature language editing prior to submission of our first manuscript and hence, we are puzzled if the quality of grammar and writing is still not satisfying. Anyway, we will check the paper again for grammar and writing before resubmission.

Major changes to the text include the following:

Changes to Figure 1 and its description:
"Each step progresses from top to bottom. Arrows outline the connections between the steps. In the first step, the time series data is aggregated appropriately (e.g., daily) and then standardized. In step II, various available distance metrics are applied to the data and investigated for multimodality. If a distance distribution is multimodal, it can be modeled by a Gaussian Mixture. This distance should be prefered for the cluster analysis in the third step. If not distance is multimodal, the framework continuous with the Euclidean distance. The cluster analysis in step III is composed of three modules. It starts with the projection in module 1and follows the structure visualization through a topographic map in module 2. This visualization enables the user to choose the number of clusters and the setting of the Boolean parameter. One arrow form module 1 bypasses module 2 and points directly to module 3, meaning that cluster analysis can be performed independently of the visualization. From module 3, an arrow points back to module 2, indicating that the number of clusters can be set as the number of visible valleys.. The clustering of the 3. Module can be further evaluated by the model of the distance distribution (last arrow between steps II and III), because using the Gaussian mixture model of step II hypothesizes that the intracluster distances of the clustering should be smaller than the Bayesian boundary defined by the Gaussian Mixture. The 3. Module can be changed to another clustering algorithm accordingly to the preference of the user. In Step IV the clustering can be validated by the topographic map. Additionally it is preferable to search for linear models and to validate the clustering externally (e.g. Heatmap). After validation in step IV, the resulting clustering of step III is used in a supervised Classification and Regression trees (CART) analysis for

the training of the un-preprocessed but aggregated data in step V. Then, the classes are defined by rules which are defined by paths in the decision tree. In the last step VI, the classes and their rules are interpreted by domain experts (c.f. [Miller, 2019]). Class-wise distribution analysis and statistical testing of relevant features can be performed to assure that explanations are tendentially contrastive (c.f. [Miller, 2019]). The details of the analytic procedure can be found in the methods sections, which is organized according to these six steps, as illustrated in the titles of the various steps in Fig. 1."

Changes to section 2.1, monitoring program:

"Data used in this work have been collected in in 2013/2014 in the Schwingbach Environmental Observatory (SEO), in central Germany. The mixed developed landscape is mainly characterized by agricultural land use (44%) and forests (48%). An in-situ hyperspectral UV-spectrometer (ProPS, Trios, Rastede, Germany, wavelength range 200–360 nm, path length 5 mm, solar panel supplied) was used to measure absorption spectra every 15 min. Prior to measurements, air blasts (5 s) were blown on the lens to prevent the optics from biofouling. Wavelengths spectra at 200–220 nm were utilized for the calculation of nitrate concentration, following a calibration with local stream water matrix (see Aubert et al. 2016 for further details). All other variables used in this machine learning approach (Table 1) were also monitored at the same high-frequency or aggregated to 15 min intervals. Discharge and stream water temperature were recorded by pressure transducers (Diver DCX, Schlumberger Water Services, ON, Canada) at two gauging stations at the outlet (q13) and upstream (q18) of the first-order stream of the Vollnkirchener Bach. Groundwater depth at three wells (GW25 hillslope, GW3 lowland, GW32 riparian zone) were measured with similar pressure transducers. Electromagnetic induction sensor (5TE) attached to EM50 data loggers (Decagon, Labcell LTD, Alton, UK) installed at 0.1 m depth in the riparian zone were used to gauge soil moisture and soil temperature. All meteorological data was collected at a climate station 4 km from the outlet (Campbell Scientific Inc., CR1000 data logger, Loughborough, UK).

In total, the dataset contains 32,196 data points for 14 different variables. Data gaps due to technical problems and data quality control reduced the available data coverage to two growing seasons (05 March 2013 to 24 September 2013, n=15,475 measurements; 27 April 2014 to 23 October 2014, n=16,721 measurements). In Table 1 abbreviations and SI units of all variables are provided. Further technical information on the SEO, the analytical procedures applied, the coding of abbreviations and the experimental design in general are outlined in detail by Aubert et al. 2016, 2017 and Orlowski et al. 2014."

What is perhaps what I find most troubling with the current work is that the key findings of the study are not new and are concepts canbe found in undergraduate level courses covering on water quality in natural systems.

We are very glad to hear that the explanations provided through the data are understandable by domain experts and have strong foundations in literature.

We would add to the introduction:

"In knowledge discovery, typically 80-90% of the knowledge acquired is already known by experts, 0-10% is incorrect due to noise in the data, and 0-10% is new knowledge (c.f. discussion in [Behnisch/Ultsch, 2015]). This remark is in line with our work because the data scientists involved in our work derived the knowledge obtained in a data-driven way without the consideration of process knowledge on water quality."

Major/Global Comments:
1) The description of the data used in the study is poor. From the introduction, it is stated that the data come from the Swingbach catchment in Germany, and from the methods section that there are 32,196

data points for 14 variables. No discussion about how many sites there, where the data were collected, or if the data even overlap temporally.

We agree with both reviewers that a subsection/supplementary is required to discuss the data collection extensively. See also our reply above on the new section of the monitoring program.

Instead of using standard terms for things like groundwater level or stream temperature, they are replaced with cryptic codes (e.g., groundwater level = GW13, GW125, or GW 132) that are make it difficult to follow the results.

We agree that some of the codes seem awkward, but they are in line with several other publications in the same catchment (Aubert and Breuer 2016, Aubert et al. 2016, Orlowski et al. 2014). Hence, we would like to keep these codes.

It took reading the previous paper to produce even a moderate understanding of what the variables are and why there appear to be duplicates. In terms of understanding underlying hydrological process, being able to discern that, for instance, that the three groundwater level measurements are for a lowland, hill slope, and riparian zone better tremendously.

We thank the reviewer for this good idea, in a revised manuscript, we will provide the mentioned table and make changes accordingly. We also included an extended description of the monitoring program that also helps to better understand the available data set.

2)   Despite having a reasonable background, in terms of both the data and the methods applied in this paper, the current text provides a poor explanation of work that is spread out throughout the text. Take for instance a very simple examination of Figure 2, which plots probability distribution estimators of "distance" for the data, as well as Gaussian end-member populations which fit the data. But what distance?

In the revised manuscript, we will change the titles of the relevant figures to Hellinger point distance and add:
"Exemplary, Figure 2 presents the probability density estimation (PDE) [Alfred Ultsch, 2005] of the distance feature $df$ of the Hellinger point distance in black and its Gaussian mixture model (GMM) in red. Specific definitions can be found in SI F."

There are 14 variables in the study, so is this a multivariate distance of some sort?

Yes. in the manuscript, we stated this in section 3.1:.
"The Hellinger distance [Rao, 1995] in the R package 'parallelDist' on CRAN was chosen." However we are grateful to the reviewer for the hint that apparently, there are two definition of the Hellinger distance in literature. Therefore, we rename Hellinger distance to Hellinger point distance in the manuscript and add a supplementary F to introduce the definition of distances in order to prevent ambiguity.

The corresponding text simply states that, "The Hellinger distance measure is selected. . . " (line 93) but Hellinger distance measures distance between distributions. So which distributions is it measuring distances between? Going back to the paper by Auger et al. (2106), they observed a tri-modal distribution of NO3 concentrations...is that what this is?

We will add a supplementary F to introduce the basic definition of distances in order to improve the understanding of Fig. 2 and prevent ambiguity:

"Let I be a finite subset of N high-dimensional points in a metric space with a distance function d(l,j), then the matrix $D = (D_{l,j})_{l,j \in I}$ is called a distance matrix of I (c.f. [Neumaier, 1981]) with each entry as $D_{l,j} = d(l,j)$ being the distance between two high-dimensional points of data. The distance matrix $D$ satisfies four conditions, meaning that the diagonal entries are all zero ($d(l,l) = 0 \ \forall \ 1 \leq l \leq N$), positive ($d(l,j) > 0 \ \forall \ l \neq j$ ), symmetric $d(l,j) = d(j,l)$ and for any l,j $d(l,j) \leq d(l,k) + d(k,j), \forall k$ (triangle inequality). Using the definition above, we define the distance feature $df$ as the upper (or lower) triangle of the symmetric distance matrix ($df = D_{lj}, \ \forall \ l > j \ , 1 \leq l \leq N, 1 \leq j \leq N$).

Given a finite dataset *I* of *N* cases, each described by *d* features, the Euclidean distance is defined as $d(l,j) = \sqrt{\sum_i (l_i - j_i)^2}$ which can be modified to the Hellinger point distance with $d(l,j) = \sqrt{\sum_i \left( \sqrt{\frac{l_i}{\sum l_i}} - \sqrt{\frac{j_i}{\sum j_i}} \right)^2}$

c.f. ([Rao, 1995; Legendre/Gallagher, 2001; Conde/Domínguez, 2018]).

For the algorithmic definition of the Hellinger point distance and the application to the data, please use the R call ?parallelDist::parallelDist() [Eckert, 2018] or inspect for details the provided source code."

In another example, we can examine how the cluster analysis is explained (note that the section title is misspelled as "Clsuter Analysis). It begins by focusing on the Pswarm method, where DataBots move data that are similar towards one another on a grid or map, but this is described using only qualitative means, such as searching for the "most potent scent" or moving towards DataBots with the most similar features. What type of similarity metric(s) is employed here?

The first module of DBS, Pswarm requires either a data matrix or a distance matrix as the input. Our manuscript states already states this in section 2.3: "During learning, each agent moves across the grid or stays in its current position in the search for the most potent scent. Hence, agents search for other agents carrying data with the most similar features to themselves with a data-driven decreasing search radius" [Thrun/Ultsch, 2020a].

Nevertheless, we will further elaborate the description of the sequence of the individual steps as follows: "Contrary to ant-based clustering algorithms, DataBots do not move data. Instead, each DataBot possesses a scent, defined by one high-dimensional data point. The equation (Eq. 18 in [Thrun/Ultsch, 2020a]) that mathematically defines the scent uses information stored in the distance matrix. No other similarity metric is used."

We will also correct the spelling mistake in the revised manuscript.

Later in the same discussion, it is explained that clustering can either be focused on compact clusters or connected clusters and t hat the authors have decided to emphasize the former (no justification given). The following text (line 14) then goes on to describe how "the choice of this parameter can be evaluated. . .". What parameter? Is this related to compact clusters or connected clusters?

With regards to the clustering, the manuscript states in chapter 2.3:

*"The clustering approach itself involves two choices. For this dataset, the compact approach is used (…). The other approach for connected structures and a general discussion of cluster structures can be found in"* [Thrun/Ultsch, 2020b]. (…) *"In praxis, the choice of this parameter can be evaluated in step IV (Fig. 1)".*

Then, in section 2.4 it is stated:
*"The clustering is valid if mountains do not partition clusters indicated by colored points of the same color and colored regions of points."*

Due to the remark of the reviewer, we will add in section 2.3:
*"In praxis, the choice of the Boolean parameter of compact versus connected can be evaluated in step IV using the topographic map as specified in Fig. 1: If a cluster is either divided in separate valleys or several clusters lie in the same valley of the topographic map, the compact (or connected) clustering approach is not appropriate for the data. An extensive discussion of this behaviour can be found in [Thrun/Ultsch, 2020b]."*

The final issue about the methods is that the reader basically must go back and forth through the paper to follow what was done and the arguments behind it. In the methods section on data preprocessing, mirrored density plots are described as being employed but no details about what they are or why they are used is provided (they look similar to violin plots but turn out to be somewhat different). However, a fuller explanation is provided much later, in the final paragraph of the methods, instead of when they are first introduced.

A manuscript on mirrored density plots is currently in its second revision, which is accessible in arxiv [Thrun et al., 2020]. The method represents a major improvement of conventional violin plots. We will revise the description of the sequence of the individual paragraphs mentioning the mirrored density plots, improve where necessary, and place this text to where mirrored density plots are referred to for the first time in our manuscript.

3) This work relies on and cites a large number of packages in R. While there's nothing fundamentally wrong with that, citing a package without describing the methods it relies on is not beneficial. My recommendation is that a new table be created which lists all the packages used and their citations. Then in text, simply state the package title and what principles or techniques the package uses.

We thank the reviewer for this very good idea, in a revised mansuscript we will provide the mentioned table and make changes accordingly.

4) The fundamental processes identified by this investigation are largely rote conclusions for scientists who study water quality in paired stream-aquifer systems. For instance, input of groundwater into a stream corresponds generally with higher temperature (due to geothermal contribution) and EC (due to longer residence time to allow for water-rock interaction). Similarly, high nitrate levels during dry days and lower stream temperature is due to a lack of dilution effect during rainfall, bur still primarily a surface water vs. groundwater contribution to streams (i.e., lower temperature). What is the value added by this analysis?

The main value added by this manuscript is the detailed model description of a new framework of how to build an explainable AI based on open-source software given environmental time series.

It may seem that some of the results obtained in this manuscript seem trivial. But the fundamental problem we see and why we think DBS is a great help for future data analysis of hydrological systems is the following: new, high-resolution hydrological *in situ* analytical instruments (like the UV hyperspectral data we used) provide impressive big data sets that are difficult to understand. Simple and classical data analysis instruments reach their limits here. The manifold relationships between the most diverse measured variables are difficult to relate to each other. Integrated methods from the field of ML can help to significantly improve these analyses. Particularly helpful in this context are those methods that start data analysis with little or no prior knowledge and can thus identify new, previously overlooked relationships. Here, DBS can close a significant gap in data analysis.

I have numerous specific comments and noted many typographical errors as well, but the feel that the items above need to be rectified before providing further feedback.

In sum, we are very grateful for all issues raised and would like to correct all the specific comments and the typographic errors that apparently still exist in the manuscript despite the acquisition of the service of springer nature for language editing. We could also provide extensive background information (e.g., define the complete DBS algorithm in Pseudocode and equations) in another supplementary w.r.t. comments discussed above if the editor states such a necessity.

[Adadi/Berrada, 2018]  **Adadi, A., & Berrada, M.**: Peeking inside the black-box: A survey on Explainable Artificial Intelligence (XAI), *IEEE Access, Vol. 6*, pp. 52138-52160. **2018.**

[Basak/Krishnapuram, 2005]  **Basak, J., & Krishnapuram, R.**: Interpretable hierarchical clustering by constructing an unsupervised decision tree, *IEEE Transactions on Knowledge and Data Engineering, Vol. 17*(1), pp. 121-132. **2005.**

[Behnisch/Ultsch, 2015]  **Behnisch, M., & Ultsch, A.**: Knowledge Discovery in Spatial Planning Data: A Concept for Cluster Understanding, *Computational Approaches for Urban Environments*, (pp. 49-75), Springer, **2015**.

[Biran/Cotton, 2017]  **Biran, O., & Cotton, C.**: Explanation and justification in machine learning: A survey, Proc. IJCAI-17 workshop on explainable AI (XAI), Vol. 8, pp. 8-13, **2017**.

[Bouveyron et al., 2012]  **Bouveyron, C., Hammer, B., & Villmann, T.**: Recent developments in clustering algorithms, Proc. ESANN, Citeseer, **2012**.

[Breiman, 2001]  **Breiman, L.**: Random forests, *Machine Learning, Vol. 45*(1), pp. 5-32. **2001.**

[Conde/Domínguez, 2018]  **Conde, A., & Domínguez, J.**: Scaling the chord and Hellinger distances in the range [0, 1]: An option to consider, *Journal of Asia-Pacific Biodiversity, Vol. 11*(1), pp. 161-166. **2018.**

[Cormack, 1971]  **Cormack, R. M.**: A review of classification, *Journal of the Royal Statistical Society. Series A (General), Vol.*, pp. 321-367. **1971.**

[Eckert, 2018]  **Eckert, A.:** parallelDist: Parallel Distance Matrix Computation using Multiple Threads (Version 0.2.4), CRAN. Retrieved from https://CRAN.R-project.org/package=parallelDist, **2018.**

[Engle et al., 2017]  **Engle, S., Whalen, S., Joshi, A., & Pollard, K. S.**: Unboxing cluster heatmaps, *BMC bioinformatics, Vol. 18*(2), pp. 63. **2017.**

[Handl et al., 2005]  **Handl, J., Knowles, J., & Kell, D. B.**: Computational cluster validation in post-genomic data analysis, *Bioinformatics, Vol. 21*(15), pp. 3201-3212. **2005.**

[Herrmann, 2011]  **Herrmann, L.**:*Swarm-Organized Topographic Mapping,* (Doctoral dissertation), Philipps-Universität Marburg, Marburg**, 2011**.

[Hewett/Leuchner, 2002]  **Hewett, R., & Leuchner, J.**: The power of second-order decision tables, Proc. Proceedings of the 2002 SIAM International Conference on Data Mining, pp. 384-399, SIAM, **2002**.

[Hofmeyr/Pavlidis, 2015]  **Hofmeyr, D., & Pavlidis, N.**: Maximum clusterability divisive clustering, Proc. 2015 IEEE Symposium Series on Computational Intelligence, pp. 780-786, IEEE, **2015**.

[Jain/Dubes, 1988]  **Jain, A. K., & Dubes, R. C.**: *Algorithms for Clustering Data*, Englewood Cliffs, New Jersey, USA, Prentice Hall College Div, ISBN: 9780130222787, **1988**.

[Johnson/Lindenstrauss, 1984]  **Johnson, W. B., & Lindenstrauss, J.**: Extensions of Lipschitz mappings into a Hilbert space, *Contemporary mathematics, Vol. 26*(1), pp. 189-206. **1984.**

[Kim et al., 2015]  **Kim, B., Shah, J. A., & Doshi-Velez, F.**: Mind the gap: A generative approach to interpretable feature selection and extraction, Proc. Advances in neural information processing systems, pp. 2260-2268, **2015**.

[Lakkaraju et al., 2016]  **Lakkaraju, H., Bach, S. H., & Leskovec, J.**: Interpretable decision sets: A joint framework for description and prediction, Proc. Proceedings of the 22nd ACM SIGKDD international conference on knowledge discovery and data mining, pp. 1675-1684, **2016**.

[Legendre/Gallagher, 2001]  **Legendre, P., & Gallagher, E. D.**: Ecologically meaningful transformations for ordination of species data, *Oecologia, Vol. 129*(2), pp. 271-280. **2001.**

[Letham et al., 2013]  **Letham, B., Rudin, C., McCormick, T. H., & Madigan, D.**: An interpretable stroke prediction model using rules and Bayesian analysis, Proc. Workshops at the Twenty-Seventh AAAI Conference on Artificial Intelligence, **2013**.

[Lipton, 2018]  **Lipton, Z. C.**: The mythos of model interpretability, *Queue, Vol. 16*(3), pp. 31-57. **2018.**

[Loyola-González et al., 2020]  **Loyola-González, O., Gutierrez-Rodríguez, A. E., Medina-Pérez, M. A., Monroy, R., Martínez-Trinidad, J. F., Carrasco-Ochoa, J. A., & García-Borroto, M.**: An Explainable Artificial Intelligence Model for Clustering Numerical Databases, *IEEE Access, Vol. 8*, pp. 52370-52384. **2020.**

[Miller, 2019]  **Miller, T.**: Explanation in artificial intelligence: Insights from the social sciences, *Artificial intelligence, Vol. 267*, pp. 1-38. **2019.**

[Miller et al., 2017]  **Miller, T., Howe, P., Sonenberg, L., & AI, E.**: Explainable AI: Beware of inmates running the asylum, Proc. International Joint Conference on Artificial Intelligence, Workshop on Explainable AI (XAI), Vol. 36, pp. 36-42, **2017**.

[Mörchen/Ultsch, 2007]  **Mörchen, F., & Ultsch, A.**: Efficient mining of understandable patterns from multivariate interval time series, *Data Mining and Knowledge Discovery, Vol. 15*(2), pp. 181-215. **2007.**

[Nauck/Kruse, 1999]  **Nauck, D., & Kruse, R.**: Obtaining interpretable fuzzy classification rules from medical data, *Artificial intelligence in medicine, Vol. 16*(2), pp. 149-169. **1999.**

[Neumaier, 1981]  **Neumaier, A.**: Combinatorial configurations in terms of distances, *Dept. of Mathematics Memorandum, Vol.*, pp. 81-09. **1981.**

[Rao, 1995]  **Rao, C.**: Use of Hellinger distance in graphical displays. Multivariate statistics and matrices in statistics, Proc. Proceedings of the 5th Tartu Conference, pp. 143-161, **1995**.

[Ribeiro et al., 2016]  **Ribeiro, M. T., Singh, S., & Guestrin, C.**: " Why should I trust you?" Explaining the predictions of any classifier, Proc. Proceedings of the 22nd ACM SIGKDD international conference on knowledge discovery and data mining, pp. 1135-1144, **2016**.

[Riid/Sarv, 2013]  **Riid, A., & Sarv, M.**: Determination of regional variants in the versification of estonian folksongs using an interpretable fuzzy rule-based classifier, Proc. 8th conference of the European Society for Fuzzy Logic and Technology (EUSFLAT-13), Atlantis Press, **2013**.

[Rousseeuw, 1987]  **Rousseeuw, P. J.**: Silhouettes: A graphical aid to the interpretation and validation of cluster analysis, *Journal of Computational and Applied Mathematics, Vol. 20*, pp. 53-65. doi https://doi.org/10.1016/0377-0427(87)90125-7, **1987.**

[Thrun, 2018]  **Thrun, M. C.**: *Projection Based Clustering through Self-Organization and Swarm Intelligence*, (Ultsch, A. & Hüllermeier, E. Eds*.,* 10.1007/978-3-658-20540-9), Doctoral dissertation, Heidelberg, Springer, ISBN: 978-3658205393, **2018**.

[Thrun et al., 2020]  **Thrun, M. C., Gehlert, T., & Ultsch, A.**: Analyzing the Fine Structure of Distributions, *PLOS ONE, preprint available at arXiv.org, Vol. in 2nd revision*, pp. arXiv:1908.06081. doi arXiv:1908.06081, **2020.**

[Thrun/Ultsch, 2020a]  **Thrun, M. C., & Ultsch, A.**: Swarm Intelligence for Self-Organized Clustering, *Artificial intelligence, Vol. in press*, pp. doi 10.1016/j.artint.2020.103237, **2020a.**

[Thrun/Ultsch, 2020b]  **Thrun, M. C., & Ultsch, A.**: Using Projection based Clustering to Find Distance and Density based Clusters in High-Dimensional Data, *Journal of Classification, Vol. accepted*, pp. doi 10.1007/s00357-020-09373-2, **2020b.**

[Ultsch, 1998]  **Ultsch, A.**: The integration of connectionist models with knowledge-based systems: hybrid systems, Proc. SMC'98 Conference Proceedings. 1998 IEEE International Conference on Systems, Man, and Cybernetics (Cat. No. 98CH36218), Vol. 2, pp. 1530-1535, IEEE, **1998**.

[Ultsch, 2005]  **Ultsch, A.**: Pareto density estimation: A density estimation for knowledge discovery, In Baier, D. & Werrnecke, K. D. (Eds.), *Innovations in classification, data science, and information systems*, (Vol. 27, pp. 91-100), Berlin, Germany, Springer, **2005**.

[Ultsch et al., 1991]  **Ultsch, A., Halmans, G., & Mantyk, R.**: CONKAT: a connectionist knowledge acquisition tool, Proc. Proceedings of the Twenty-Fourth Annual Hawaii International Conference on System Sciences, Vol. 1, pp. 507-513, IEEE, **1991**.

[Ultsch/Korus, 1995]  **Ultsch, A., & Korus, D.**: Integration of neural networks and knowledge-based systems, Proc. IEEE Int. Conf. Neural Networks, Perth, Australia, **1995**.

[Ultsch et al., 1995]  **Ultsch, A., Korus, D., & Kleine, T.**: Integration of neural networks and knowledge-based systems in medicine, Proc. Conference on Artificial Intelligence in Medicine in Europe, pp. 425-426, Springer, **1995**.

[Weinstein, 2008]  **Weinstein, J. N.**: A postgenomic visual icon, *Science, Vol. 319*(5871), pp. 1772-1773. **2008.**

[Wilkinson/Friendly, 2012]  **Wilkinson, L., & Friendly, M.**: The history of the cluster heat map, *The American Statistician, Vol.*, pp., **2012.**